# On the Pitfalls of Using the Residual Error as Anomaly Score

**Felix Meissen**[1,2]                                                FELIX.MEISSEN@TUM.DE
[1] *Technical University of Munich, Germany*
[2] *Klinikum Rechts der Isar, Munich, Germany*
**Benedikt Wiestler**[2]                                             B.WIESTLER@TUM.DE
**Georgios Kaissis**[1,2,3]                                          G.KAISSIS@TUM.DE
[3] *Imperial College London, UK*
**Daniel Rueckert**[1,2,3]                                           DANIEL.RUECKERT@TUM.DE

**Editors:** Under Review for MIDL 2022

## Abstract

Many current state-of-the-art methods for anomaly localization in medical images rely on calculating a residual image between a potentially anomalous input image and its ("healthy") reconstruction. As the reconstruction of the unseen anomalous region should be erroneous, this yields large residuals as a score to detect anomalies in medical images. However, this assumption does not take into account residuals resulting from imperfect reconstructions of the machine learning models used. Such errors can easily overshadow residuals of interest and therefore strongly question the use of residual images as scoring function. Our work explores this fundamental problem of residual images in detail. We theoretically define the problem and thoroughly evaluate the influence of intensity and texture of anomalies against the effect of imperfect reconstructions in a series of experiments. Code and experiments are available under https://github.com/FeliMe/residual-score-pitfalls.
**Keywords:** Anomaly Segmentation, Semi-supervised Learning

## 1. Introduction

Semi-supervised anomaly segmentation has shown great successes in the last years, for example in industrial defect detection (Bergmann et al., 2019). Acquiring labels in medical images is both expensive (labeling oftentimes has to be performed by trained doctors) and inherently noisy due to inter-rater variability. Detecting pathologies in images without the need for labels would consequently be of great value in the medical domain, which led to increased research interest in this direction. In large medical 3D scans, a simple per-scan binary detection is not enough, as it would still require a professional to localize the anomaly. Ergo, accurate localization of pathologies is a necessity for clinical applicability in this domain. Usually, anomaly segmentation approaches for medical images train a generative model to learn the distribution of "healthy" images without visible pathologies. After training on such data, the model will likely fail to generate parts of the image that contain (out-of-distribution) pathologies and replace it with "healthy" (in-distribution) anatomy instead. In many cases, the pixel-wise residual between the original and the input image is used as a score to detect anomalous regions in the image. However, the results of Meissen et al. (2021) suggest that the approaches using this principle so far have only proven to be "white object detection" methods. On the other hand, unsupervised anomaly segmentation has potentially many applications in the medical domain where the pixel intensities of

anomalies are similar to those of healthy pixels, such as colonoscopy images (Tian et al., 2021), Chest X-rays, or ultrasound images (Tan et al., 2021). Having anomaly segmentation methods for this kind of data would consequently be of great value. The results of the Medical Out-of-Distribution Analysis Challenge 2020 (Zimmerer et al., 2020) also show that anomaly segmentation does not produce clinically usable results yet.

**Contribution** In this work, we theoretically and experimentally evaluate the use of the pixel-wise residual error as a score for detecting anomalous regions in grayscale images. We simulate anomalies with specific characteristics on brain MR images and study the influence of anomaly texture, size, and their contrast to the surrounding tissue on the detection performance. We identify blind spots of this scoring metric and their causes.

## 2. Background and Related Work

The approaches for pixel-wise anomaly segmentation using residuals in medical grayscale images can be divided into two main categories:

**Reconstruction-based Methods** An Autoencoder-like model, trained on "healthy" data only, computes a reconstruction $\hat{\mathbf{x}}$ from an input image $\mathbf{x} \in \mathcal{X} \subset \mathbb{R}^{H \times W}$. The anomaly map $\mathbf{a}$ is the pixel-wise residual between $\mathbf{x}$ and $\hat{\mathbf{x}}$

$$a_i = |x_i - \hat{x}_i|, \forall i \in H \times W, \tag{1}$$

where $\hat{\mathbf{x}}$ can also be an approximation of an expectation as in variational Autoencoders. Pawlowski et al. (2018) trained different Autoencoders – including a denoising Autoencoder and a variational Autoencoder – to detect anomalous regions in the form of tumors in CT images of the human brain. Baur et al. (2020a) used a UNet-like Autoencoder to detect MS lesions and glioblastoma in brain MRI. Their model produces higher-fidelity reconstructions than vanilla- or variational Autoencoders. However, they acknowledge that their method only works for hyperintense anomalies.

**Restoration-based Methods** use a generative model $g$ – such as a VAE or a GAN – that have a smooth, interpolable latent space $\mathcal{Z}$, such that latent vectors $\mathbf{z} \in \mathcal{Z}$ that are close in the latent space are also close in the image space $\mathcal{X}$. For a query image $\mathbf{x}$, they iteratively move along the latent manifold to find a generated image $\hat{\mathbf{x}} = g(\mathbf{z})$ that is close to $\mathbf{x}$, but is still in the learned normative distribution. Restoration finds

$$\underset{\mathbf{z}}{\arg\min} \, \mathcal{L}\left(g(\mathbf{z}), \hat{\mathbf{x}}\right) \tag{2}$$

via gradient descent. This method was originally proposed by Schlegl et al. (2017) with a Generative Adversarial Network (GAN). You et al. (2019) implemented another restoration variant that directly optimizes the pixels of the input image $\mathbf{x}$ to maximize the ELBO of a VAE or a Gaussian Mixture VAE. Restoration-based approaches use iterative updating and are therefore orders of magnitude slower than reconstruction-based methods. Step size, number of optimization steps, and regularization-choice, and strength are additional hyper-parameters that need to be considered when using restoration.

For a more thorough investigation into the aforementioned categories, we refer the reader to the comparative study of Baur et al. (2021). Lastly, Meissen et al. (2021) did not train

a machine learning model at all but used the histogram equalized input images directly as anomaly maps without computing a residual. This method by design can only detect hyperintense anomalies, but outperformed most competing methods on several benchmark datasets for lesion and tumor detection in brain MRI. Their results suggest that the competing methods themselves are merely "white object detection" methods that flag any hyperintense part of the image as anomalous.

## 3. Preliminaries and Notation

Assume we have a generative model $g$ that has learned the normative distribution perfectly, and will always generate "healthy" anatomy only.

Let $\mathbf{x} \in \mathcal{X} \subset \mathbb{R}^{H \times W}$ be a query image with an anomalous region indexed by $h$. Generative machine learning models can in practice not reconstruct images perfectly. Even for in-distribution images $g(\mathbf{x})_i = \hat{x}_i = x_i + e_i$ the error $e_i$ is not uniformly distributed over all pixels, but is larger at some parts of the image (i.e. at edges).

Let $\tilde{\mathbf{x}}$ be the ground truth "healthy" query image. The magnitude of the anomaly map at pixels $i \in h$ is bounded by $|a_i| \leq |\tilde{x}_i - (x_i + e_i)|$. Consequently, if for any pixel $j$ outside of the anomaly $h$, $|e_j| > |\tilde{x}_i - (x_i + e_i)|, \forall i \in h$, then errors will be introduced.

## 4. Experiments

In this section, we present a series of experiments that investigate the limitations of using the pixel-wise residual error as a score to segment anomalies in grayscale images.

**Data** We perform our experiments on the publicly available brain MRI data of the MOOD Analysis Challenge 2020 (Zimmerer et al., 2020). It consists of 800 T2-weighted scans of healthy young adults with $256 \times 256 \times 256$ voxels per scan. The scans have voxel-intensities normalized between 0 and 1, and the average intensity of non-zero voxels (voxels that are not part of the background) is 0.446. We split the data into a training and a test dataset, containing 60% and 40% of the scans respectively. We slice each scan into 2D images along the axial direction and select the 20 slices around the center for the training dataset and the center slice for the test dataset. For every image $\tilde{\mathbf{x}}$, we create its anomalous counterpart $\mathbf{x}$ by sampling a circular patch, $h$ with a radius of 20 pixels, at a random location inside the brain and altering the pixel values $x_i, \forall i \in h$.

**Evaluation Metric** In all our experiments, we use the pixel-wise average precision (AP) averaged over all images as the evaluation metric. AP is equivalent to the area under the precision-recall curve and is suited for measuring anomaly segmentation performance because the anomaly maps aren't discrete segmentations, but continuous heatmaps.

### 4.1. Experiment 1 — Evaluating the Influence of Anomaly Intensity

In the first experiment, we take a random healthy image $\tilde{\mathbf{x}}$ from the test dataset. To create the anomalous image $\mathbf{x}$, we sample $h$, as described in Section 4, and replace the pixel values in $h$ with an intensity $I$ between 0 and 1. We simulate an Autoencoder that can perfectly remove the anomaly from $\mathbf{x}$, but generates imperfect reconstructions $\hat{\mathbf{x}}$ by applying Gaussian blur with $\sigma$ between 0 and 5 to the healthy image: $\hat{\mathbf{x}} = g(\tilde{\mathbf{x}}, \sigma)$. We compute the pixel-wise

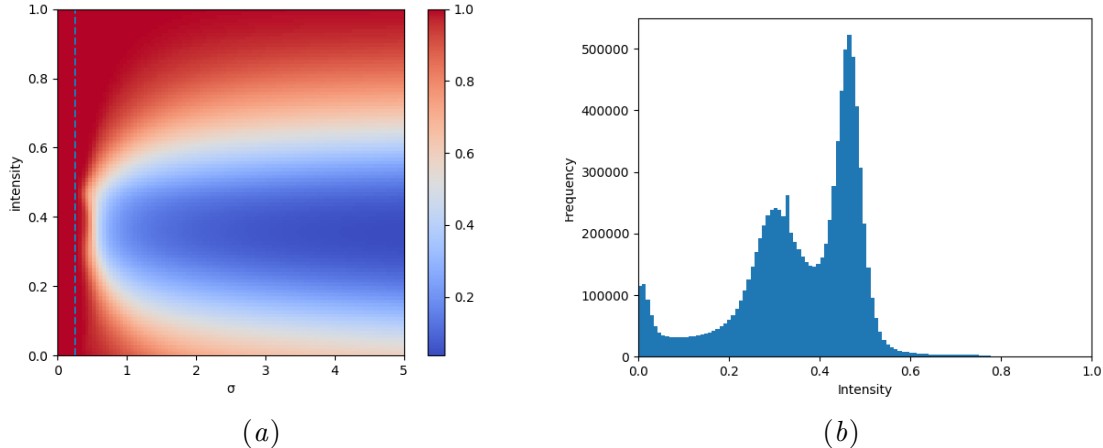

$(a)$ $(b)$

Figure 1: (a) Average precision of imperfect reconstructions at varying intensities with a vertical dashed line at $\sigma = 0.25$. (b) Histogram of pixel-intensities over the test dataset without anomalies.

anomaly map according to Equation (1). This experiment is repeated for every image in the test dataset. The averaged results are shown in Figure 1($a$).

**Findings** Average precision decreases heavily (already at small $\sigma$-values) when the anomalies have intensities between 0.2 and 0.6. This indicates a large blind spot of the residual-based anomaly segmentation around the average pixel intensity of the object. The higher average precision for anomalies with high intensities can partly be explained by the histogram in Figure 1($b$). Only a few pixels in the test dataset have values $> 0.6$, so the residual is likely to be high for these anomalies. Note that Gaussian blur starts having a noticeable effect on pixels when $2\sigma \geq 0.5$. Otherwise, more than 95.45% of the Gaussian probability mass are on the center pixel, so little to no blurring occurs.

### 4.2. Experiment 2 — Evaluating the Influence of Anomaly Texture

After examining the effect of the intensity of the anomalous regions in experiment 1, the following experiment will investigate the effect of their texture. Just like in experiment 1, we are simulating a generator with imperfect reconstructions using Gaussian blurring of the healthy image: $\hat{\mathbf{x}} = g(\tilde{\mathbf{x}}, \sigma)$. We again fix the anomaly size $h_s$ to 20 pixels and use three types of artificial anomalies that change the texture of the anomalous region but apply at most minimal changes to the overall intensity:

- **Sink deformation** (Tan et al., 2020): Pixels in the anomalous region $h$ with size $h_s$ get shifted away from the center of the region $h_c$. In the new image $\hat{\mathbf{x}}$, the pixel at index $J$ is replaced by the pixel from the original image $\tilde{\mathbf{x}}$ at index $V$: $\hat{\mathbf{x}}_J = \tilde{\mathbf{x}}_V$, $\forall J \in h$, where $J = (i, j)$, $V = J + (1 - s)(J - h_c)$, and $s = \left( \frac{||J - h_c||_2}{h_s} \right)$.

- **Source deformation** (Tan et al., 2020): Analogous to sink deformation, but pixels get shifted towards the center, so $V = h_c + s(J - h_c)$.

- **Pixel shuffle**: The anomalous image is generated by randomly permuting all pixel indices in $h$.

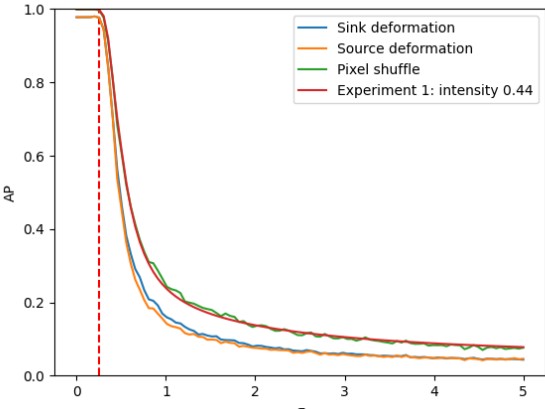

Figure 2: Average precision vs Gaussian blur strength $\sigma$ for sink deformation, source deformation, pixel shuffle, and the curve with intensity 0.44 from experiment 1. The vertical dashed bar indicates $\sigma = 0.25$.

**Findings**  The results in Figure 2 show a lower performance for sink- and source deformation. The results for the pixel shuffle anomaly align closely to that of using a constant intensity of $I = 0.44$ from experiment 1. Note that this value roughly equals the average intensity of brain pixels in the test dataset. These results indicate that the texture of the anomaly is not important for anomaly segmentation using the reconstruction error. Even when an extreme texture difference is applied, the performance is equivalent to applying no texture at all.

### 4.3. Experiment 3 — Transferring to Generative Machine Learning Models

Now that we described the AP landscape in experiments 1 and 2, experiment 3 will test the validity of the assumption that a generative model can be simulated by applying Gaussian blurring to the healthy image $\tilde{\mathbf{x}}$. For this experiment, we train multiple generative models on the training dataset. We train vanilla and spatial Autoencoders (Baur et al., 2019) with varying latent space dimensions, and a hierarchical VQ-VAE (Razavi et al., 2019) which is known to produce high-quality reconstructions. The vanilla and spatial Autoencoders all share the same encoder and decoder architecture with five layers. The VQ-VAE uses the original architecture and, therefore, has the largest latent space with $64 \times 32 \times 32 + 64 \times 64 \times 64$ dimensions. Architecture- and Training details can be found in the supplementary material.

### 4.3.1. EXPERIMENT 3.1

For the first sub-experiment, we generate the anomalous ($\mathbf{x}$) and the healthy image ($\tilde{\mathbf{x}}$) from the test dataset analogous to experiment 1. We use the trained model $g$ to generate a reconstruction of the **healthy** image $\hat{\mathbf{x}} = g(\tilde{\mathbf{x}})$, and again compute the anomaly map according to Equation (1). This setup again assumes that the generator only produces "healthy" anatomy, but with reconstruction imperfections. The results are depicted in Figure 3($a$). In addition to the AP curves of the single models, the Figure also contains the curves with the best matching $\sigma$ from experiment 1 for the VQ-VAE, and the vanilla Autoencoder with $\mathcal{Z} \subset \mathbb{R}^{128}$. The best matching curves are the ones with the least absolute error to the AP curve of the respective model.

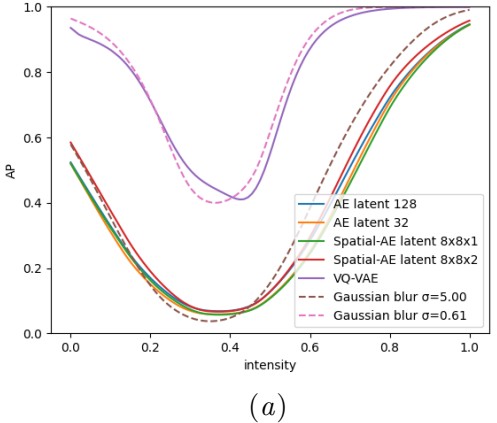 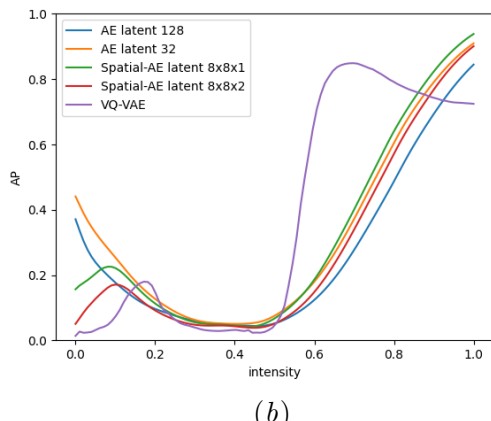

$(a)$ $(b)$

Figure 3: Average precision vs intensity of the anomalous pixels for three vanilla Autoencoders and a VQ-VAE. (a) Reconstructing the **healthy** image, and includes AP curves of best matching $\sigma$ from experiment 1. (b) Reconstructing the **anomalous** image.

**Findings** The experiment produces curves similar to the ones from experiment 1, both showing a significant drop in performance at intensities around $I \approx 0.4$. The performance of the VQ-VAE is similar to blurring the input image with $\sigma = 0.61$, and the performance of the vanilla Autoencoders to blurring with $\sigma = 5.0$. The Autoencoders with larger latent space (more capacity) produce better reconstructions and, thus, have a slightly better performance in this experiment.

### 4.3.2. EXPERIMENT 3.2

The second sub-experiment is performed analogous to the first one, with the difference that $\hat{\mathbf{x}}$ is the reconstruction of the **anomalous** image $\hat{\mathbf{x}} = g(\mathbf{x})$. This setup is more realistic since it does not rely on the assumption that the generator produces only "healthy" images.

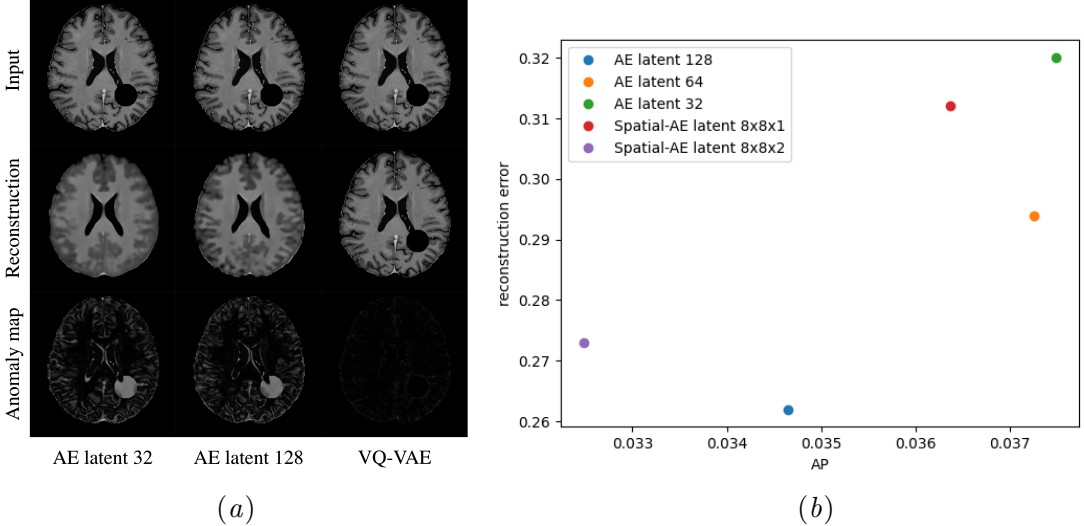

Figure 4: (a) Input images, reconstructions and anomaly maps for three different models. (b) Reconstruction error vs average precision for different models.

**Findings** The results in Figure 3(b) show a reversed ordering of the AP-curves for vanilla Autoencoders. Here, models with larger latent space perform worse. The VQ-VAE drops most dramatically in performance compared to experiment 3.1. While its AP is still high for high-intensity anomalies, it drops fast and is especially low in the area of lower intensities. Figure 4(a) explains this behavior. Models with lower reconstruction error tend to also reconstruct anomalies. This is well pronounced in the VQ-VAE, but can already be observed in the Vanilla AE where the reconstructed image of the larger model (latent space dimension of 128) has a slightly enlarged left lateral ventricle in the ventral aspect. Figure 4(b) shows that models with a lower reconstruction error on the test dataset tend to show inferior anomaly segmentation performance.

The spatial Autoencoders and the VQ-VAE show a slight performance increase in Figure 3(b) around the intensity $I = 0.1$. We hypothesize that this is due to the pixel probability being low around this intensity in the test dataset as shown in Figure 1(b). Thus, the residual is likely to be higher in this region. The same hypothesis explains the strong performance of all models on high-intensity anomalies.

## 5. Discussion

Above, we performed a series of experiments to investigate the influence of certain anomaly characteristics (such as intensity and texture) on the performance of using the pixel-wise reconstruction error as a score to segment anomalies in medical grayscale images. Our work uncovers severe problems with this method.

In Section 4.1, we discovered a large blind spot for anomalies with intensities around 0.2 to 0.6, which corresponds roughly to the intensity range that is well covered by the pixels in

the object as shown in Figure 1(b), ergo in the area where the probability of finding $j$, such that $|e_j| > |\tilde{x}_i - (x_i + e_i)|$, is high. Considering this, the blind spot might be even larger for data that covers a broader intensity range. Additional experiments in this direction can be found in the Appendix. We further found in Section 4.2 that the important characteristic for anomalies to be detected is their intensity. Their texture, however, is irrelevant. While we found in experiment 3.1 (Section 4.3.1) that the assumption to simulate a machine learning generator with a Gaussian blurring process is reasonable, Section 4.3.2 made clear that performance can not trivially be increased by generating higher-quality images. In fact, increasing the model's reconstruction quality leads to anomalies being reconstructed as well, which further hinders their detection and argues against this simple strategy. The same behavior was analyzed in the comparative study of Baur et al. (2021). They even found a positive correlation between the reconstruction error of normal pixels and the segmentation performance. Overall, our findings are in-line with current research by Meissen et al. (2021) who found that machine learning models that use pixel-wise residuals as anomaly maps are simple "white object detection" methods. This work further provides an explanation for why anomaly segmentation in brain MRI was so far mainly successful in imaging sequences where the anomalies of interest are known to be hyperintense (Pawlowski et al., 2018; Atlason et al., 2019; Zimmerer et al., 2019; Chen and Konukoglu, 2018; Pinaya et al., 2021; Baur et al., 2020b). Since the anomalous pixels there would be primarily located on the right-hand side of the histogram in Figure 1(b), the probability mass of normal pixels with a similar intensity is lower, and the residual error is, therefore, expected to be higher. Histograms supporting this claim can be found in the Appendix. However, as the type and therefore also the characteristics of potential pathologies are unknown beforehand, our selection of artificial anomalies is not representative for all real-world cases. For example, abnormal size of anatomical structure could be considered as an anomaly. Nevertheless, our experiments allow to draw conclusions about all pathologies that are expressed via intensity- or texture difference.

## 6. Conclusion and Future Research

This work has shown that using the pixel-wise residuals between a grayscale image and its reconstruction to localize anomalies has severe limitations. Especially anomalies with intensities that are well covered by other pixels in the object can not reliably be detected with this method, regardless of their texture. While the main reason for this phenomenon is imperfect reconstructions of the generator model, increasing reconstruction quality does not trivially increase performance, but oftentimes leads to anomalies being reconstructed as well, hindering their detection. The successes of these methods in industrial defect detection suggest that the problem is less severe in RGB images. In future work, we will therefore research how medical scans with multiple channels can be constructed and their effect on anomaly detection. Also, advanced post-processing of the residual maps as in Muñoz-Ramírez et al. (2021) can potentially alleviate this problem and presents an interesting direction for further research. Lately, self-supervised methods that use artificial anomalies for training became popular (Tan et al., 2020, 2021) and need further investigation. Lastly, promising research directions are likelihood-based models and methods based on latent space features, such as Bergmann et al. (2020), to obtain pixel-wise likelihood estimates.

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

## Appendix A. Samples of Anomalies and Residual Maps

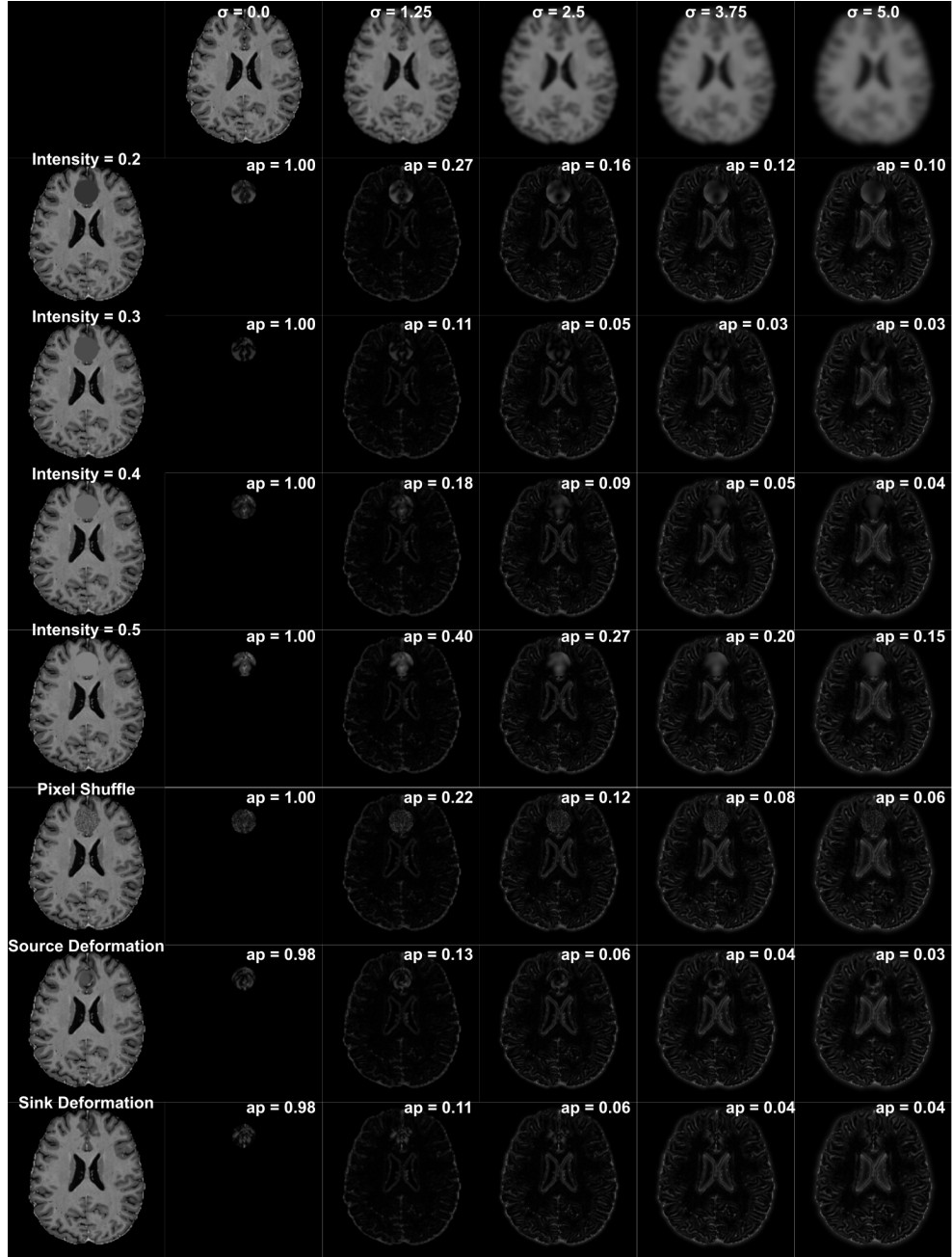

Figure 5: Samples of artificial anomalies, the blurred input images and the corresponding residual maps from experiments 1 and 2.

## Appendix B. Architecture and Training Details

Here we describe the architectures of the machine learning models used in our experiments.

### B.1. Model Architectures

**Vanilla Autoencoder**   This model has a classical encoder-decoder structure with a latent bottleneck. The encoder consists of five downsampling layers, each with a convolutional layer with stride two and same padding, followed by batch normalization and a leaky ReLU activation function. The output of the encoder has the shape $c \times 8 \times 8$ and is flattened before a fully connected layer transforms it into the latent vector $\mathbf{z} \in \mathcal{Z}$ of size $c_l$. From there, another fully connected layer and a reshape operation transforms ‡ back to $c \times 8 \times 8$ dimensions which is fed through the decoder. The decoder mirrors the encoder but uses Transpose Convolutions instead of regular ones. Lastly, a final Convolution is applied to the output of the decoder to generate the final reconstruction.

**Spatial Autoencoder**   The architecture of the spatial Autoencoder is analogous to the one of the vanilla Autoencoder. The only difference is the bottleneck where now a Convolution transforms the output of the decoder into $\mathbf{z} \in \mathcal{Z}$, and another Convolution transforms this vector back to fit into the decoder. The latent space in these models is of shape $c_l \times 8 \times 8$ and retains an image-like structure.

**VQ-VAE**   This model uses the original architecture proposed in (Razavi et al., 2019) for image resolutions of $256 \times 256$. Apart from the bottleneck, this model differs from the others by using residual blocks in the encoder and decoder and by having two latent spaces: one at resolution $64 \times 64 \times 64$, and one at resolution $64 \times 32 \times 32$, making it the largest latent space of all models considered.

### B.2. Training details

We implemented all models using PyTorch (Paszke et al., 2019). All models are trained using the AdamW optimizer (Loshchilov and Hutter, 2017) with a learning rate of 0.001 for 10000 steps with a batch size of 128 to minimize the L1 loss between the input image and its reconstruction. During training, we use 5% of the training dataset for validation. When training the VQ-VAE, we weighted the latent loss by a factor of 0.25 compared to the reconstruction error.

## Appendix C. Experiment 1 with Histogram Equalization

Here, we perform an additional experiment, investigating the hypothesis from Section 5 that the blind-spot of the residual error might be larger when the data covers a broader intensity range. For this, we repeat experiment 1 with histogram-equalized data. Specifically, we perform histogram equalization to each image slice in the test dataset, and only to pixels in the image that belong to the brain.

The results, alongside the histogram of the equalized data can be found in Figure 6. The performance drop is now distributed symmetrically over the whole intensity range, confirming our initial hypothesis.

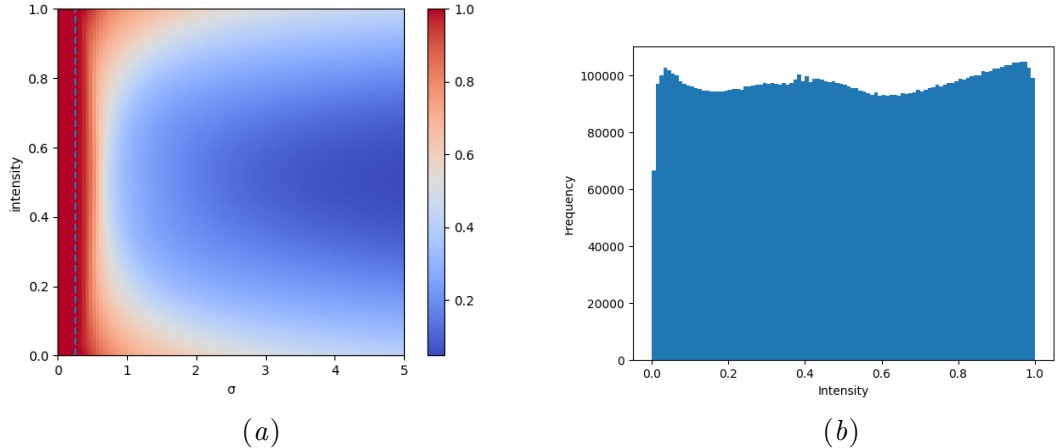

Figure 6: (a) Average precision of imperfect reconstructions at varying intensities with a vertical dashed line at $\sigma = 0.25$. (b) Histogram of pixel-intensities over the histogram-equalized test dataset.

## Appendix D. Histograms of Brain MRI Datasets

Here, we show the histograms of normal and anomal voxels of four data sets commonly used to evaluate anomaly segmentation for brain MRI. In all data sets the intensities of anomal voxels tend to be higher then the intensities of normal voxels.

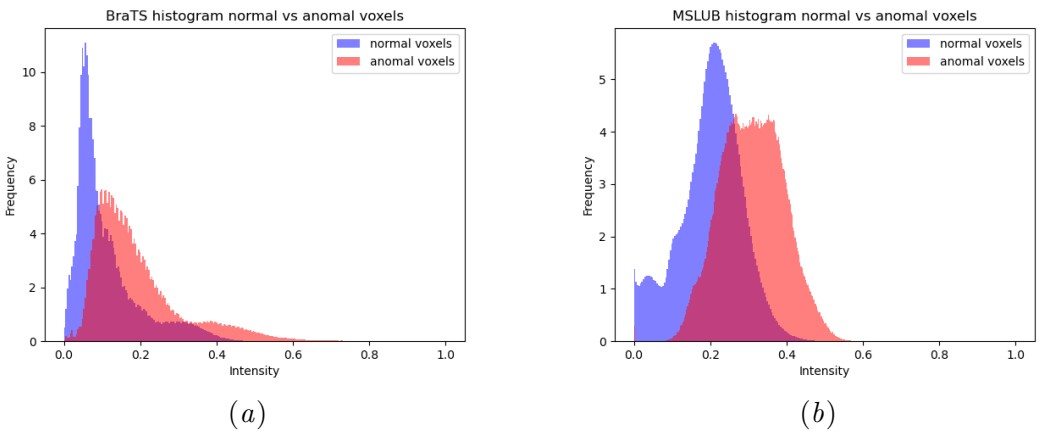

Figure 7: Histograms of normal and anomal pixels for (a) BraTS (Menze et al., 2015) and (b) MSLUB (Lesjak et al., 2018) FLAIR images.

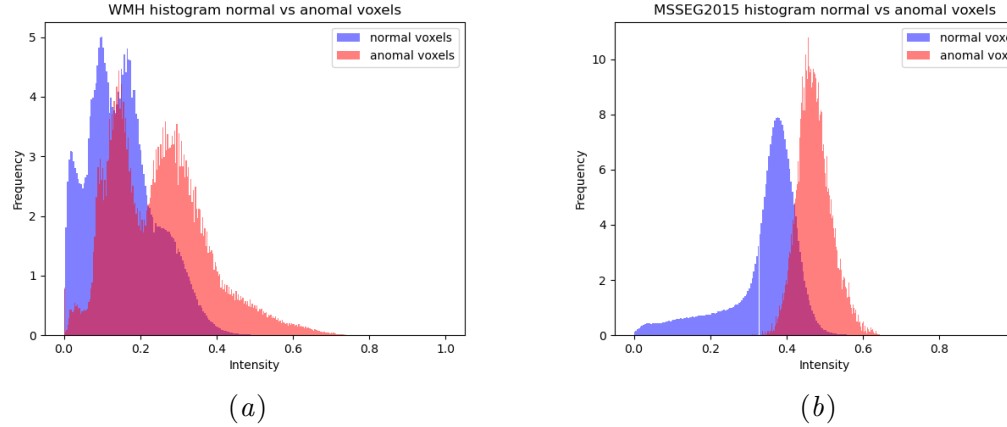

Figure 8: Histograms of normal and anomal pixels for (a) WMH (Kuijf et al., 2019) and (b) MSSEG2015 (Carass et al., 2017) FLAIR images.

