# OpenReview forum: "On the Pitfalls of Using the Residual as Anomaly Score"
_MIDL.io/2022/Conference — MIDL 2022_

### Official Review · Reviewer_peL8 · 2022-01-21

**Confidence:** 5
**Preliminary Rating:** 5
**Recommendation:** Best Paper Award, Poster

**Summary:**

Authors present a thorough study in which they demonstrate that the residual error cannot be reliably used in anomaly detection applications. There is increasing interest in anomaly detection applied to medical images, but the literature published so far has some serious limitations. One of them is that most methods are simply white-object-detectors (demonstrated by these same authors last MICCAI) and that using the reconstruction/restoration/residual error is not a good way to find anomalies. Both issues (white objects, residual error) are interrelated, because it appears that using the residual does work if-and-only-if you are looking for hyperintense (and large) anomalies. But that is of course not solely the case in real medical imaging applications.

This work conclusively demonstrates the limitations associated with using the residual in medical image anomaly detection.

**Strengths:**

The paper is very clear and well written. Authors use both simulations as well as actual implementations of generative models to demonstrate their point. They address an important limitation of anomaly detection, which is itself of increasing interest in the field.

**Weaknesses:**

There are no major weaknesses with this work. I have some minor suggestions:
- sections 4.1 and 4.2 could use some Figures that show how the generated anomalous images look like.
- the Discussion could address the other strategy to identify anomalies: based on the features in the latent space.

**Deanonymize Review:**

no

**Detailed Comments:**

Caption of Figure 2: 4.44 should probably be 0.44 .

**Final Rating After The Rebuttal:**

5: Strong Accept

**Justification Of The Final Rating:**

I am happy with the comments of the authors.                                                                                                                                                                     q

**Paper Type:**

validation/application paper

**Questions To Address In The Rebuttal:**

The work itself is very clear, so I have no particular questions that need to be addressed in this work. If there is some space left, authors could consider describing the alternative solution to identify anomalies: based on the latent space. While that is not the focus of this work, it would make the work a bit more complete.

**Special Issue:**

yes

---

### Official Review · Reviewer_JeX8 · 2022-01-24

**Confidence:** 5
**Preliminary Rating:** 3
**Recommendation:** Poster

**Summary:**

The main idea of this paper is to explore the limits of anomaly detection methods based on unsupervised generative models trained on healthy images only. The key assumption of such models is that reconstructed image from any pathological input data will contain healthy looking tissue at the pathological localisations, so that residual image corresponding to the difference between the original and reconstructed images will outline anomalous localisations. The authors advocate that such methods are limited by the level of noise in the reconstructed image that may hinder the detection of the anomaly, depending on its intensity. This study is based on three series of experiments, the first two series contain “simulated” reconstructed images with varying noise level or textures, as well as a third series of experiment validating the use of simulated images by comparing them to reconstructed images generated from vanilla and VQ-VAE autoencoders.

**Strengths:**

-This study tackles a hot topic in the community. Progress in the field of anomaly detection models based on residual maps (derived from generative autoencoders) is closely followed in the community since they allow relaxing the constraints in terms of annotated data.

-The paper is well structured with a well conducted experimental study based on simulated data first, allowing to evaluate the impact of the different parameters of interest, ie noise level and texture.


**Weaknesses:**

Conclusion of the paper should be moderated. Indeed, the proposed study has some limitations that should be discussed:
-The metric used for the evaluation is the image-wise average precision (AP) which is computed at the voxel level, where all voxels are considered individually without accounting for their neighboors, unlike visual analysis. I suggest the authors to perform some postprocessing of the heatmaps, as usually done, including clustering, and cleaning of the cluster maps based on size or localization priors. As the authors mention, residual error is not uniformly distributed, so that false positive clusters might be localized to specific regions (borders) and easily removed based on simple erosion or dilation masks. I guess this is likely to significantly clean the heatmaps from small false positive clusters and improve the performance metric.
 -To add to the previous point, the authors should also discuss more advanced post-processing methods that accounts for normalisation of the residual errors based on its distribution in the control population as, proposed in [Munoz-Ramirez, MICCAI-MLCN21] for instance.

-the paper is lacking illustrations of the synthetic images, especially regarding the synthetic anomalous data. Providing example images for the 3 experiments will help the reader visualizing the influence of the different parameters, including Gaussian noise level or texture type on of the visual detection task.


**Deanonymize Review:**

no

**Detailed Comments:**

-Reference to your arxiv paper (Meissen 2021) is incomplete, I guess this paper was presented at the 2021 Brain Lesion segmentation challenge.

**Final Rating After The Rebuttal:**

3: Borderline

**Justification Of The Final Rating:**

I thank the authors for their review and added illustrations in the revised version of the paper. I maintain my rating to ‘borderline’. The paper would really gain soundness by adding some post-processing of the residual maps. This might indeed moderate the conclusion of the paper.

**Paper Type:**

both

**Questions To Address In The Rebuttal:**

-The authors should address the points reported in the 'weakness' section, especially regarding the discussion of the limitation of the proposed study as well as provide illustrations of the synthetic data to help evaluate the difficulty of the segmentation task.

**Special Issue:**

no

---

### Official Review · Reviewer_ruGs · 2022-01-27

**Confidence:** 3
**Preliminary Rating:** 3
**Recommendation:** Poster

**Summary:**

* The paper is a discussion on the effectiveness of pixel-wise residual errors for localizing anomalies in grayscale medical images.

* With well-designed simulation experiments, the authors show that pixel-wise residues stem from two factors:
    * the presence of anomalous regions in the test image
    * imperfect reconstructions of generative models

* It is shown that these two factors may interfere with one another, leading to incorrect anomaly localization - especially when anomalous regions contain intensities similar to those of healthy pixels.

* Overall, the paper raises a valid and important discussion for the unsupervised anomaly detection community, but some points may have been overstated (explained below).

**Strengths:**

* The paper raises an important question for the unsupervised anomaly detection community.
* The simulation experiments are well-designed.
* The writing is fairly clear, and the paper is easy to understand. Thank you!

**Weaknesses:**

* While the observations are indeed interesting, I am a bit concerned if they are sufficient to justify a publication at this conference. The paper would have been much stronger if the observations would have been complemented with a proposal of an alternate method.

* I believe that some conclusions may need to be toned down.
    * It seems to me that the experiments show the limitations of pixel-wise residual maps, and not residual maps in general. This should be clarified.
    * It is claimed that generative models that provide sharp reconstructions of healthy images are also capable of reconstructing anomalous regions. This seems to have been based on the observation from the VQ-VAE model in Fig 4a. I don't believe this is sufficient to make a general statement about all generative models. Authors should either cite relevant papers to support this statement, or tone down the statement.




**Deanonymize Review:**

no

**Detailed Comments:**

* I would refrain from calling Section 3 'Theoretical analysis'. It seems to me that this section introduces the notation, and then explains in that notation what has been discussed in words before.

* The description of sink and source deformations in unclear. What is $V$? What value of $h_s$ is used in the experiments?

* Visual examples of texture-based anomalies and corresponding residual maps will help. Are these anomalies realistic? It is fine if they are not, but this should be acknowledged, and the validity of the experiment despite this fact should be explained.

* I believe the paper may also benefit from visualizations of  simulated intensity-based anomalies and corresponding residual maps in Experiment 1.

* Caption of Figure 2 should say intensity $0.44$ instead of $4.44$?

**Final Rating After The Rebuttal:**

4: Weak Accept

**Justification Of The Final Rating:**

The paper presents an interesting question for the unsupervised anomaly detection community, and backs this with well-designed simulated experiments. In the rebuttal, several points that were unclear in the original version have been clarified, and some excessive claims have been adequately moderated.

**Paper Type:**

validation/application paper

**Questions To Address In The Rebuttal:**

* The experiments show problems with usage of pixel-wise residual maps. Is it possible that alternate ways of using residual maps may be suitable for localizing anomalies? Instead of aggregating pixel-wise residuals, does the paper suggest that the community should focus on developing methods to quantify structural patterns in residual images? For instance, is it possible that the residual maps for texture-based anomalies show the texture of the anomaly, but also contain other regions due to imperfect reconstructions. Can the authors provide visualizations of such residual maps?

* Can the authors provide examples of real anomalies that have intensities similar to those of healthy image pixels? If so, do any papers in the anomaly detection literature consider such anomalies? A discussion around this could make the motivation for this paper a lot stronger.

**Special Issue:**

no

---

### Meta-Review · Area_Chair_NkDj · 2022-02-13

**Recommendation:** Accept (Poster)
**Confidence:** 4

**Metareview:**

I thank all reviewers for their time and effort spent reviewing this paper and their engagement with the rebuttal process. I also thank the authors for their detailed and extensive rebuttal and the changes made to the manuscript, which in my opinion adequately addressed the reviewers’ concerns. On balance there is sufficient support to warrant acceptance of the paper. I look forward to seeing it presented at MIDL.

---

### Decision · Program_Chairs · 2022-02-28

Accept